# *Rv1453* is associated with clofazimine resistance in *Mycobacterium tuberculosis*

Lei Zhang,[1] Ye Zhang,[1] Yuanyuan Li,[1] Fengmin Huo,[2] Xi Chen,[1] Hui Zhu,[1] Shaochen Guo,[1] Lei Fu,[1] Bin Wang,[1] Yu Lu[1]

**ABSTRACT** Clofazimine (CFZ) has been repurposed for treating tuberculosis (TB), especially multidrug-resistant tuberculosis (MDR-TB). However, the mechanisms of resistance to clofazimine are poorly understood. We previously reported a mutation located in the intergenic region of *Rv1453* that was linked to resistance to CFZ and demonstrated that an *Rv1453* knockout resulted in an increased minimum inhibitory concentration (MIC) of CFZ. The current study aims to go back and describe in detail how the mutation was identified and further explore its association with CFZ resistance by testing additional 30 isolates. We investigated MICs of clofazimine against 100 clinical strains isolated from MDR-TB patients by microplate alamarBlue assay. Whole-genome sequencing (WGS) was performed on 11 clofazimine-resistant and 7 clofazimine-susceptible strains, including H37Rv. Among the 11 clofazimine-resistant mutants subjected to WGS, the rate of mutation in the intergenic region of the *Rv1453* gene was 55% (6/11) in clofazimine-resistant strains. Among another 30 clofazimine-resistant clinical isolates, 27 had mutations in the intergenic region of the *Rv1453* gene. A mutation in the *Rv1453* gene associated with clofazimine resistance was identified, which shed light on the mechanisms of action and resistance of clofazimine.

**IMPORTANCE** Tuberculosis (TB) is an infectious disease caused by the bacterium *Mycobacterium tuberculosis*, especially the emergence of multidrug-resistant tuberculosis (MDR-TB) brings great distress to humans. Clofazimine (CFZ) plays an important role in the treatment of MDR-TB. To understand the underlying mechanism of clofazimine resistance better, in this study, we review and detail the findings of the mutation of intergenic region of *Rv1453* and find additional evidence that this mutation is related to clofazimine resistance in 30 additional isolates. The significance of our research is to contribute to a comprehensive understanding of clofazimine-resistant mechanisms, which is critical for reducing the emergence of resistance and for anti-TB drug discovery.

**KEYWORDS** *Mycobacterium tuberculosis*, clofazimine, *Rv1453*, drug resistant

The problem of tuberculosis (TB) resistance has become more severe owing to the increasing incidence of multidrug-resistant tuberculosis (MDR-TB) and extensively drug-resistant tuberculosis (XDR-TB). The latest data reported in 2022 showed that the success rate for people treated for TB in 2020 was 86%; however, the treatment success rate for MDR-TB was 60% worldwide (1). Because of the difficulty in developing new drugs, it is essential to slow the development of resistance to currently available drugs. Identifying the genes responsible for drug resistance in TB is important not only for contributing to the molecular diagnosis of drug-resistant TB and explaining how drug resistance acts, but also for developing next-generation drugs to combat resistance.

Clofazimine (CFZ) was one of the first riminophenazine antibiotics developed and was initially described in 1957 (2). Although this drug was ignored for decades, the emergence of MDR- and XDR-TB has revived interest in CFZ as an anti-TB agent and some

Address correspondence to Yu Lu, luyu4876@hotmail.com.

Lei Zhang, Ye Zhang, and Yuanyuan Li contributed equally to this article. Author order was determined by workload.

The authors declare no conflict of interest.

See the funding table on p. 9.

studies have confirmed that CFZ has good anti-TB activity (3, 4). CFZ is recommended as a group B agent in the 2018 WHO treatment guidelines for MDR-TB and rifampicin-resistant (RR)-TB and is playing an increasingly critical role in the treatment of drug-resistant TB. Compared with other anti-TB drugs, the adverse effects of CFZ are generally minor and rarely life-threatening, and CFZ exhibits bactericidal activity even after the cessation of treatment (5, 6). In addition, the cost and resistance rate of CFZ are low (7), although the resistance rate of CFZ may be higher than expected because of its widespread use in recent years.

Mutations in the *Rv0678*, *Rv1979c*, and *pepQ* genes are the previously described mechanisms of *Mycobacterium tuberculosis* resistance to CFZ. *atpE* is the target of bedaquiline (BDQ) and, thus, is not involved in CFZ resistance. The significance of *Rv1979c* mutations remains unclear (8). *Rv0678* and *pepQ* genes are related to the cross-resistance of CFZ and BDQ, which causes an increase of two to four times in the minimum inhibitory concentration (MIC) of CFZ (9, 10). In addition, Zhang et al. (11) and Xu et al. (12) found that mutations in *pepQ* were also associated with CFZ resistance, although the changes in MIC were modest, with increases lower than fourfold. However, the MIC of CFZ against the *M. tuberculosis* isolates from one patient was increased, but no mutations were found in *Rv0678*, *Rv1979c*, and *pepQ* and their respective upstream regions (13), suggesting the existence of other unknown mechanisms of CFZ resistance. Therefore, the identification of new CFZ resistance-related genes will contribute to a comprehensive understanding of CFZ-resistant mechanisms that is critical for reducing the emergence of resistance and for antiTB drug discovery.

We have reported that the transcription factor Rv1453 mediates CFZ resistance in *M. tuberculosis* by regulating the expression of the *qor* gene (14). The research was based on the present study, in which we screened the whole-genome sequences of 11 CFZ-resistant *M. tuberculosis* strains selected from 100 multidrug-resistant clinical isolates and found that a mutation in the intergenic region of the *Rv1453* gene was associated with resistance to CFZ in *M. tuberculosis*. Hence, this study reports the discovery of mutation in the intergenic region of *Rv1453* gene and provides 11 CFZ-resistant *M. tuberculosis* strains in more detail, and validates mutation in the intergenic region of *Rv1453* gene in a further 30 CFZ-resistant clinical isolates which not belongs to 100 MDR-TB isolates. We suggested that the mutation in the intergenic region of *Rv1453* gene contributes to CFZ resistance in *M. tuberculosis* by altering the transcriptional and translational levels of its target gene. This study provides further evidence that the *Rv1453* gene and CFZ resistance are related.

## MATERIALS AND METHODS

### Bacterial strains and culture conditions

All clinical strains were selected from TB strains preserved in Beijing Chest Hospital. 100 MDR-TB clinical strains were selected from MDR-TB strains between January 2014 and November 2016. 30 CFZ-resistant clinical isolates to validated the *Rv1453* intergenic region mutation were selected between March 2013 and May 2016. Clinical strains of our laboratory preserved were selected between June 2012 and December 2013. *M. tuberculosis* H37Rv (ATCC 27294) was grown in 7H9 liquid medium supplemented with 0.2% glycerin, 0.05% Tween 80, and 10% oleic acid-albumin-dextrose-catalase (OADC) at 37°C for about 14 days (exponential phase), or in 7H10 solid medium with 0.5% glycerin and 10% OADC at 37°C for about 4 weeks (exponential phase). The strain growth conditions were also appropriate for clinically isolated strains and *Rv1453* gene knockout strain.

### MIC determination

MICs were determined by the microplate AlamarBlue assay (MABA), and the details have been described previously (15). Briefly, the final concentrations of CFZ, TBI-166, and BDQ

ranged from 10 to 0.005 mg/L. Then added 100 μL bacteria ($2 \times 10^5$ CFU) to wells. The plates were incubated at 37℃. After day 7 of incubation, added 12.5 μL of 20% Tween 80 and 20 μL of alamarBlue to plates. After incubation at 37℃ for another 24 h, the fluorescence was measured at an excitation wavelength of 530 nm and an emission wavelength of 590 nm. The MIC was defined as the lowest concentration eliciting a reduction in fluorescence of ≥90% relative to the mean fluorescence of replicate drug-free controls. The susceptible reference strain *M. tuberculosis* H37Rv was included in each batch as a control and the MIC value of CFZ to H37Rv strain was 0.11 mg/L.

## Identification of CFZ-resistant strains

*M. tuberculosis* H37Rv and all clinically isolated strains were cultured to exponential phase, MIC of each strain to CFZ was determined by MABA method, and drug-resistant strains were screened according to the critical concentration (CC) of CFZ. The CC of CFZ was determined to be 1.2 mg/L based on a previous study (12). The CC of TBI-166 was preliminarily determined as 0.6 mg/L, which is a 10-fold MIC value of TBI-166 against strain H37Rv. The CC of BDQ was determined as 0.5 mg/L based on the previous study (16). MIC assays were performed in three independent experiments.

## Genomic DNA extraction

Genomic DNA was extracted from freshly cultured bacteria by using the Wizard Genomic DNA Purification Kit (Promega) according to the manufacturer's protocol.

## Whole-genome sequencing

The genomic DNA samples from 11 CFZ-resistant isolates, 6 CFZ-susceptible isolates, and *M. tuberculosis* H37Rv were sequenced using the Illumina Hiseq sequencing platform. Paired-end sequencing libraries for genomic DNA of each strain were constructed using SPARK DNA Sample Prep Kit (enzymatics) according to the manufacturer's instructions. $Q$ value was used to evaluate sequencing quality: $Q = -10\log10(E)$, $E$ is the sequencing error rate. Bwa software (version 0.7.12) was used for Genome mapping. Reads were aligned with the reference sequence of *M. tuberculosis* H37Rv (NC_00962.3), and the sequencing depth is 370-fold to 740-fold. The maximum-likelihood phylogenetic tree was constructed by FastTree, based on SNPs from whole-genome sequences and was annotated by tvBOT (17).

## PCR and DNA sequencing

PCR and sequencing of the *Rv1453* gene were subjected to PCR amplification using the primers (forward primers: 5′-CGACGCCAACAACTACGAAC-3′; reverse primers: 5′-GGCCGGCAACGTAACTCAGG-3′) and the genomic DNA of CFZ-resistant clinical isolates selected as a template. The amplification was performed as follows: 94℃ for 5 min, followed by 30 cycles of 94℃ for 30 s, 60℃ for 30 s, 72℃ for 40 s, and a final extension of 10 min at 72℃. The obtained PCR products were sent for sequencing to determine the mutations harbored in *Rv1453* gene in the isolated CFZ-resistant mutants.

## RESULTS

### Sensitivity of clinical isolates to CFZ

We measured the MICs of CFZ in clinical strains isolated from MDR-TB patients by MABA, and 1.2 mg/L was determined as the critical concentration of CFZ. Seven CFZ-resistant strains were selected from 100 MDR-TB isolates. The resistance rate of CFZ was 7%. We identified another four CFZ-resistant strains among the preserved strains in our laboratory by MABA. A total of 11 CFZ-resistant strains were obtained, and the CFZ MIC values ranged from 1.221 to 3.168 mg/L (Table 1). For seven CFZ-susceptible strains, the CFZ MIC values ranged from 0.11 to 0.59 mg/L.

**TABLE 1** Data of 11 CFZ-resistant clinical isolates[b]

| ID | Date of isolation | CFZ (mg/L) | TBI-166 | BDQ | SRA | Lineage[a] | Rv0678 | Rv1979c | pepQ | Rv1453 intergenic region | Medication administration record | Drug resistance profile | CFZ treatment | Preserved isolates |
|---|---|---|---|---|---|---|---|---|---|---|---|---|---|---|
| 11492 | 201212 | 3.618 | + | + | SRR22873596 | 2.2.2 | WT | WT | WT | WT | INH, RIF, EMB, PZA, SM, KM, PAS, CAP, PA, MFX, PTO | INH, RIF, EMB, SM, PAS, CPM, AM | Yes | Yes |
| 12657 | 201307 | 1.221 | + | − | SRR22873595 | 2.2.1 | S53L | WT | WT | C-T | INH, RIF, CLR, AMX/CLV, EMB, LVFX, AM | INH, RIF, EMB, SM, OFX, LVFX | No | Yes |
| 13476 | 201312 | 2.244 | + | + | SRR22873594 | 2.2.1 | L117R | WT | WT | WT | INH, RIF, EMB, PZA, RFP, LVFX | INH, RIF | No | Yes |
| 13480 | 201312 | 1.279 | − | − | SRR22873609 | 2.2.1 | WT | V52G | WT | WT | INH, RIF, PZA, AM, PTO, LVFX, RFP, EMB, MFX, CLR | INH, RIF, EMB, OFX, LVFX, PAS | No | Yes |
| 13908 | 201401 | 2.471 | − | − | SRR22873608 | 2.2.2 | A59V | WT | WT | WT | AMX/CLV, PAS, RFB, INH, RIF, PZA, EMB, AM, LVFX, PA, RFP, PTO | INH, RIF, EMB, OFX, LVFX, PAS | No | No |
| 14470 | 201403 | 1.596 | + | − | SRR22873607 | 2.2.1 | WT | WT | WT | C-T | INH, RIF, EMB, PTO, PAS, MFX, CLR, AMX/CLV, PZA, AM | INH, RIF, EMB, SM, PAS, AM | Yes | No |
| 14907 | 201407 | 2.744 | + | + | SRR22873606 | 2.2.1 | WT | WT | WT | WT | CAP, RIF, PA, RFP, EMB, PZA, PTO, GFX, CLR, AMX/CLV, MFX, PAS, CS | INH, RIF, SM, OFX, LVFX, PAS, CPM, AM | Yes | No |
| 15171 | 201409 | 1.234 | − | − | SRR22873605 | 2.2.1 | WT | V52G | WT | C-T | INH, RIF, PZA, AM, PTO, LVFX, RFP, EMB, MFX, CLR | INH, RIF, EMB, SM, OFX, LVFX, PAS | No | No |
| 17584 | 201507 | 3.229 | − | + | SRR22873604 | 2.2.1 | Frame shift (T141 insertion) | WT | WT | C-T | AMX/CLV, PAS, RFB, RIF, PA, RFP, EMB, PZA, PTO, GFX CLR, AMX/CLV, MFX | INH, RIF, EMB | Yes | No |
| 17851 | 201509 | 1.754 | + | − | SRR22873603 | 2.2.1 | WT | WT | WT | C-T | PAS, CS, CAP, LZD, INH, RIF, PZA, EMB, SM, AM, PAS, LVFX, MFX, CAP, PTO | INH, RIF, SM, OFX, LVFX, PAS, CPM, AM | Yes | No |
| 17908 | 201509 | 1.806 | + | − | SRR22873602 | 2.2.1 | WT | WT | WT | C-T | CS, CLR, AMX/CLV, INH, RIF, PZA, EMB, MFX, PZA, PTO, CAP, PAS | INH, RIF, EMB, SM, OFX, LVFX, PAS, CPM, AM | No | No |

[a]Identification of M. tuberculosis isolates lineages was based on the previous study (18).
[b]AM, amikacin; AMX/CLV, amoxicillin/clavulanate; BDQ, bedaquiline; CAP, capreomycin; CFZ, clofazimine; CLR, clarithromycin; CS, cycloserine; EMB, ethambutol; GFX, gatifloxacin; INH, isoniazid; KM, kanamycin; LVFX, levofloxacin; LZD, linezolid; MFX, moxifloxacin; OFX, ofloxacin; PA, isoniazid aminosalicylate; PAS, para-aminosalicylate; PTO, protionamide; PZA, pyrazinamide; RFB, rifabutin; RFP, rifapentine; RIF, rifampicin; SM, streptomycin; TBI-166, riminophenazine; WT, wild type; +, resistant; −, susceptible.

## Genes associated with CFZ resistance in *M. tuberculosis* identified by whole-genome sequencing

Illumina paired-end DNA re-sequencing was performed on DNA samples from 11 CFZ-resistant strains and 7 CFZ-susceptible strains (6 CFZ clinically susceptible strains and 1 as the *M. tuberculosis* H37Rv strain as the control). The proportion of per base quality higher than 20 was not less than 90%. A total of 532 SNV mutant genes and 121 mutation sites in the intergenic region were obtained by removing the detected mutation sites from seven susceptible CFZ strains. None of the 532 mutated genes had been reported to confer resistance to other antiTB drugs.

We analyzed the frequencies of any mutation (whether previously reported or not) detected in the *Rv0678*, *Rv1979c*, and *pepQ* genes of drug-resistant and susceptible CFZ strains, and found that only 4 of the 11 drug-resistant CFZ strains had *Rv0678* gene mutations, with a mutation rate of 36% [1 with S53L, 1 with L117R, 1 with A59V, and 1 with a frameshift (T141 insertion)]. We evaluated mutations in the *Rv1979c* gene and found that for only one site in it (V52G), the mutation rate was 18% (2/11). There were no strains with simultaneous mutations of *Rv1979c* and *Rv0678*. No *pepQ* mutation was found in any strains that were sequenced (Table 1). The mutation rate of these three genes in CFZ-resistant clinical strains is low, which may not explain all the resistance of CFZ.

Whole-genome sequence comparison of the 11 CFZ-resistant clinical strains and 7 CFZ-susceptible strains revealed a gene potentially associated with CFZ resistance. According to the calculation of mutation sites in intergenic regions, six CFZ-resistant strains had a mutation in the 17th base pair upstream of the *Rv1453* gene coding region and this mutation was not present in seven CFZ-susceptible strains. The mutation frequency of the *Rv1453* intergenic region was more than 50% (Table 1).

Based on medication administration records, we found that the clinical isolates were resistant to CFZ regardless of whether the patient had used CFZ (5/11) or not (6/11). However, all these clinical-resistant strains were isolated from patients who had used antiTB drugs (Table 2). The four strains with the *Rv0678* gene mutation and two strains with the *Rv1979c* gene mutation were isolated from patients who had not taken CFZ. There were no *pepQ* mutations in these CFZ-resistant strains, and these strains were isolated from patients who had not taken BDQ. In addition, mutations of the *Rv1453* intergenic region could also occur in strains isolated from patients who had not taken CFZ. All 11 resistant strains were resistant to isoniazid and rifampicin, more than half of the strains were resistant to ethambutol, streptomycin, ofloxacin, levofloxacin, *para*-aminosalicylate, and amikacin, and patients who had taken CFZ were resistant to at least one of the fluoroquinolones or second-line injections.

## Phylogenetic analysis of CFZ-resistant isolates

Maximum-likelihood phylogenetic trees were created using SNPs from the whole-genome sequencing (WGS) of the 11 CFZ-resistant isolates, 6 CFZ-susceptible isolates,

**TABLE 2** PCR validation of C-17T mutation in the *Rv1453* gene

| Isolate | Presence of mutation | Isolate | Presence of mutation | Isolate | Presence of mutation |
|---------|---------------------|---------|---------------------|---------|---------------------|
| 11819 | Yes | 19531 | Yes | 19682 | Yes |
| 12458 | Yes | 19552 | Yes | 19683 | Yes |
| 12881 | Yes | 19562 | Yes | 19762 | No |
| 14522 | Yes | 19575 | Yes | 19821 | Yes |
| 14916 | Yes | 19579 | No | 19872 | Yes |
| 17402 | Yes | 19623 | Yes | 19905 | Yes |
| 19405 | Yes | 19624 | Yes | 19960 | Yes |
| 19477 | No | 19629 | Yes | 19994 | Yes |
| 19524 | Yes | 19655 | Yes | 20053 | Yes |
| 19526 | Yes | 19681 | Yes | 20155 | Yes |

and the strain H37Rv to investigate the molecular evolution and genetic diversity of the CFZ-resistant strains. The scale corresponds to genetic distance. The phylogenetic tree showed that of the six CFZ-resistant isolates containing the *Rv1453* C-17T mutation, strains 12657, 17851, and 17908 formed a single clade. There is evidence of homoplasy, and the *Rv1453* mutation has independently evolved four times (Fig. 1).

## Validation of the *Rv1453* intergenic region mutation in CFZ-resistant clinical strains

In addition to the 100 clinical drug-resistant isolates that were screened, we expanded the screening scope and obtained 30 CFZ-resistant clinical isolates. We conducted PCR validation on whether these drug-resistant isolates had a C-17T mutation in the *Rv1453* intergenic region. First, we extracted the whole genomic DNA of the validation strains, and then we designed PCR primers based on the gene sequences of the C-17T mutation

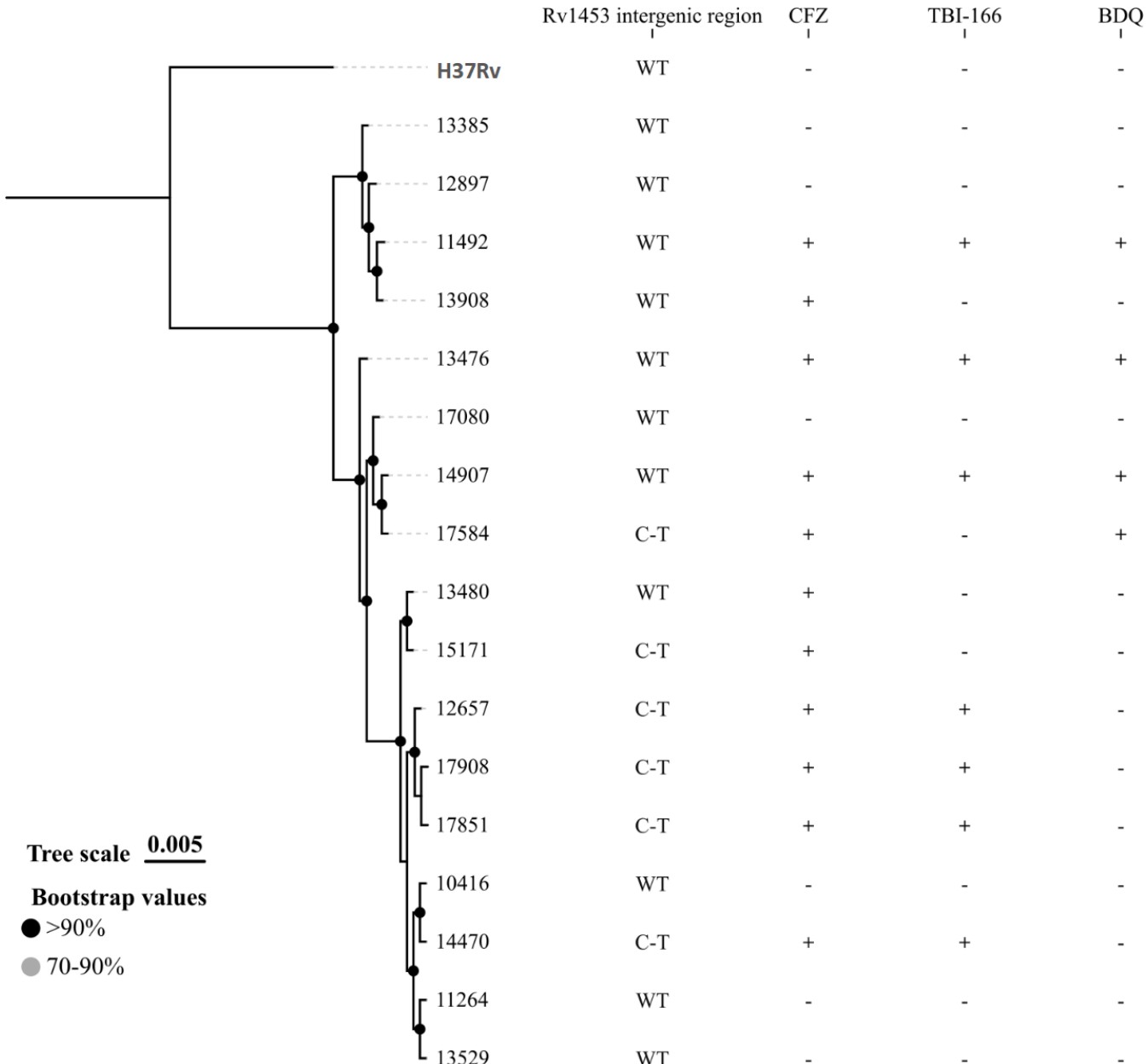

| | Rv1453 intergenic region | CFZ | TBI-166 | BDQ |
|---|---|---|---|---|
| H37Rv | WT | - | - | - |
| 13385 | WT | - | - | - |
| 12897 | WT | - | - | - |
| 11492 | WT | + | + | + |
| 13908 | WT | + | - | - |
| 13476 | WT | + | + | + |
| 17080 | WT | - | - | - |
| 14907 | WT | + | + | + |
| 17584 | C-T | + | - | + |
| 13480 | WT | + | - | - |
| 15171 | C-T | + | - | - |
| 12657 | C-T | + | + | - |
| 17908 | C-T | + | + | - |
| 17851 | C-T | + | + | - |
| 10416 | WT | - | - | - |
| 14470 | C-T | + | + | - |
| 11264 | WT | - | - | - |
| 13529 | WT | - | - | - |

Tree scale 0.005

**Bootstrap values**
● >90%
● 70-90%

**FIG 1** Phylogenetic analysis of 18 *M. tuberculosis* isolates. +, resistant; –, susceptible.

in the intergenic region of the *Rv1453* gene. We used the DNA of these strains as the template to amplify and sequence the target fragments by PCR to capture the intergenic region. The mutations of the corresponding sites of each strain were compared according to the determination results. The results showed that 27 strains had the C-17T mutation in the intergenic region of the *Rv1453* gene, which was frequently found in CFZ-resistant strains, and the mutation rate was 90% (Table 2).

## DISCUSSION

Because the drug resistance rate of CFZ was very low in clinical practice before the widespread use of CFZ, there have been few studies of CFZ-resistant clinical strains of *M. tuberculosis*. In this study, we examined CFZ-resistant clinical strains of *M. tuberculosis* isolated from MDR-TB patients rather than laboratory-induced strains. Compared with single drug-induced resistance strains, gene mutations in clinical isolates occur under pressure from multiple drugs and are also related to irregular drug use, which is more reflective of clinical practice. Although the background of clinical isolates is more complex, research on clinical isolates may have greater value in guiding clinical practice.

The mechanism of action of CFZ is still undefined, although some studies have shown that the mechanism of action of CFZ involves the following two processes. First, the oxidation of reduced CFZ, which can lead to the generation of antimicrobial reactive oxygen species (9, 19–21). Second, the almost complete inhibition of potassium uptake by CFZ through lysophosphatidic-mediated membrane dysfunction, which can reduce the amount of ATP by interfering with the membrane potential of *M. tuberculosis* (19, 21, 22). The resistance rate of CFZ may be low because it has several antibacterial mechanisms. CFZ is playing an increasingly important role in the treatment of drug-resistant TB. To slow down the emergence of drug resistance arising from the increasingly widespread use of CFZ, it is necessary to study the mechanism of drug resistance further.

*Rv0678* is an important genetic locus involved in resistance because it is found in both laboratory and clinical settings (23). The rate of *Rv0678* gene mutation was reported as 6.3% among 347 clinical strains isolated from MDR-TB patients, whereas the rate was only 0.7% in clinical strains isolated from non-MDR-TB patients (24). We obtained 11 CFZ-resistant strains from MDR-TB clinical strains, and *Rv0678* gene mutations with different mutation sites and a frameshift were found in four CFZ-resistant strains. Dozens of different *Rv0678* mutations have been identified, and the 193 and 466 sites may be mutation hot spots in this gene (9, 25, 26). Although the site of resistance mutation is not identical to the previously reported site, mutations at different sites in the *Rv0678* gene affect the protein structure by altering the amino acid sequence, and these structural alterations are expected to be involved in CFZ resistance by increasing efflux pump expression. The level of resistance to CFZ in the *Rv0678* mutant strain is low, suggesting that the *Rv0678* gene mutation may be not the main cause of CFZ resistance. *Rv1979c* encodes a possible permease that might be involved in the transport or uptake of CFZ (11). In the present study, *Rv1979c* gene mutations were found in two CFZ-resistant strains with the same mutation site. However, mutations in the *Rv1979c* gene were not found in CFZ-resistant strains in other studies (13, 27), suggesting that the role of mutations in the *Rv1979c* gene in CFZ-resistant *M. tuberculosis* strains needs to be investigated further. *pepQ* encodes a putative proline-specific aminopeptidase (10), but the mechanism by which *pepQ* mutations result in reduced susceptibility to CFZ is unclear. No mutation was detected in *pepQ* among CFZ-resistant isolates in this study. This suggests the importance of expanding our understanding of CFZ resistance mechanisms by identifying additional mutations.

Although mutations in *Rv0678*, *Rv1979c*, and *pepQ* confer previously described resistance to CFZ, the mechanism of resistance to CFZ for some of the isolates identified *in vitro* and in clinical settings remains unknown. It is likely that there are other unidentified mechanisms of resistance to CFZ. Here, we identified a mutation in the intergenic region of *Rv1453* (C-17T) that was associated with CFZ resistance. In validation experiments, 27 out of 30 CFZ-resistant clinical strains had the C-17T mutation in the

intergenic region of the *Rv1453* gene. This result supported the conclusion that the C-17T mutation in the intergenic region of the *Rv1453* gene was related to CFZ resistance. The MIC of CFZ against the *Rv1453* gene knockout strain was four times higher than that of *M. tuberculosis* H37Rv strain (14).

Increasing numbers of transcription factors associated with *M. tuberculosis* resistance have been investigated, including *Rv0678*, *Rv1909c* (28), *OhrR* (29), *RbpA* (30), *Rv3143* (31), and *Rv0324* (32). Without exception, these transcription factors can mediate drug resistance in *M. tuberculosis* by regulating the expression of their corresponding target genes. This also suggests an important role for transcription factors in drug resistance in *M. tuberculosis*. The total length of the *Rv1453* gene is 1266 bp and is adjacent to the *qor* gene (Fig. 2). The Rv1453 protein, which is approximately 46.6 kDa in size, is mainly found in the cell wall and plasma membrane of *M. tuberculosis*, has a helix-turn-helix domain at the C-terminus, and is structurally similar to the PucR transcriptional regulatory protein of *Bacillus subtilis*. We previously determined that the *Rv1453* gene is associated with CFZ resistance in *M. tuberculosis* by constructing *Rv1453* knockout strain and testing the CFZ MIC values. Rv1453 protein can bind to the *qor* gene and inhibit the expression of quinone oxidoreductase (QOR) (14). QOR in *M. tuberculosis* catalyzes the two-electron reduction of quinone to hydroquinone to prevent the generation of free radicals during redox, and transfers reducing equivalents to the electron transport chain, which is vital for energy production (33). We speculate that the C-17T mutation in the *Rv1453* intergenic, downregulated the expression of the *Rv1453* gene. This affects the expression of QOR, which changes the menanquinone/menaquinol ratio and inhibits the reduction reaction of CFZ by type-II NADH dehydrogenase, leading to *M. tuberculosis* resistance to CFZ. We need to construct *Rv1453* (C-17T) mutation strain in the further study. What is more, the MIC value of CFZ and the expression of *Rv1453* gene in *Rv1453* (C-17T) mutation strain should be detected. The MIC results from this study indicated that *Rv1453* may confer cross-resistance to TBI-166.

Although CFZ had MICs greater than 1.2 mg/L against all 11 CFZ-resistant clinical isolates, the MIC of CFZ against the *Rv1453* gene knockout strain was four times higher than that of *M. tuberculosis* H37Rv and was not higher than 1.2 mg/L. This suggests that the high MICs of CFZ-resistant clinical strains may be due to a combination of multiple resistance genes. Of the six *Rv1453* gene mutant strains isolated in this study, three of the six (50%) patients had taken CFZ, whereas the other three of the six (50%) patients had not. Thus, the mutations in the *Rv1453* gene did not depend on whether CFZ had been taken. Periodic monitoring of CFZ resistance is particularly important in patients without a history of CFZ treatment and drug sensitivity testing is necessary to prevent adverse events caused by unnecessary drug administration before deciding to treat MDR-TB patients with CFZ. Although isolates 11492 and 14907 were resistant to CFZ, no mutation sites were found in the *Rv0678*, *Rv1979c*, *pepQ*, and *Rv1453* genes, and the cause of resistance in these isolates should be determined in additional research.

The limitations of our study should also be discussed. First, the main limitation of our study is the small number of CFZ-resistant and CFZ-susceptible isolates analyzed, particularly by WGS. We expect that more clinical strains can be obtained in the future by screening in different regions. Second, more information about transcriptomes and Rv1453 protein structure will help us understand the transcriptional regulation of the

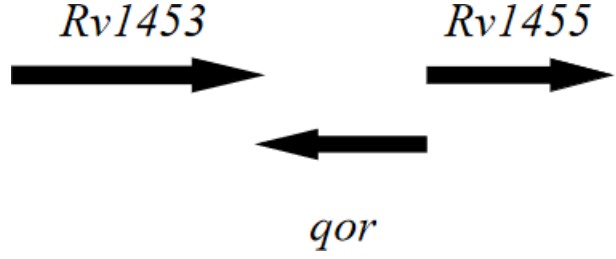

**FIG 2** Location of *Rv1453* and its adjacent genes.

*Rv1453* gene and extend our knowledge of the mechanism of resistance in *M. tuberculosis* to CFZ and other drugs.

## Conclusion

We review and describe in detail the identification of mutation in the intergenic region of *Rv1453* gene with a high mutation frequency (55%) involved in CFZ resistance. Importantly, further investigation of its association with CFZ resistance through testing of additional isolates. We hypothesize that the mutation in the intergenic region of *Rv1453* gene contributes to CFZ resistance in *M. tuberculosis* by downregulating *Rv1453* gene expression. Our study offers new insights and valuable information that will contribute to the rapid identification of CFZ-resistant isolates in a clinical setting and shed new light on the mechanisms of action of CFZ and CFZ resistance.

### ACKNOWLEDGMENTS

This work was supported by the National Natural Science Foundation of China (81973367), Beijing Hospitals Authority Clinical Medicine Development of Special Funding Support (ZYLX202123), and Beijing Municipal Administration of Hospitals' Ascent Plan (DFL20221402).

Lei Zhang and Yuanyuan Li carried out the data collection and interpretation, and drafted the manuscript. Ye Zhang participated in the design of the study and performed the experiments. Fengmin Huo participated in strain collection. Xi Chen and Hui Zhu participated in the data collection. Shaochen Guo assisted with statistical analysis. Lei Fu and Bin Wang assisted with experiments. Yu Lu participated in the design and conceived the experiments, read and revised the manuscript, and supervised the study. All authors read and approved the final manuscript.

### AUTHOR AFFILIATIONS

[1]Department of Pharmacology, Beijing Key Laboratory of Drug Resistance Tuberculosis Research, Beijing Chest Hospital, Capital Medical University, Beijing Tuberculosis and Thoracic Tumor Research Institute, Beijing, China
[2]National Clinical Laboratory on Tuberculosis, Beijing Key Laboratory for Drug-Resistant Tuberculosis Research, Beijing Chest Hospital, Capital Medical University, Beijing Tuberculosis and Thoracic Tumor Institute, Beijing, China

### AUTHOR ORCIDs

Yu Lu http://orcid.org/0000-0002-9038-1884

### FUNDING

| Funder | Grant(s) | Author(s) |
| --- | --- | --- |
| MOST \| National Natural Science Foundation of China (NSFC) | 81973367 | Yu Lu |
| Beijing Hospitals Authority Clinical Medicine Development of Special Funding Support | ZYLX202123 | Yu Lu |
| Beijing Municipal Administration of Hospitals' Ascent Plan | DFL20221402 | Yu Lu |

### DATA AVAILABILITY

The WGS data were submitted to the Sequence Read Archive of the National Center for Biotechnology Information (PRJNA914083).

### ETHICS APPROVAL

As the study only concerned laboratory testing of mycobacteria without the direct involvement of human subjects, ethics approval was not sought.

## ADDITIONAL FILES

The following material is available online.

## Supplemental Material

**Tables S1 to S2 (Table S1 to S2.docx).** Details of MTB strains.

## Open Peer Review

**PEER REVIEW HISTORY (review-history.pdf).** An accounting of the reviewer comments and feedback.

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
