## [Reviewer comments · Microbiology Spectrum]

Microbiology Spectrum

Rv1453 is associated with clofazimine resistance in Mycobacterium tuberculosis

Lei Zhang, Ye Zhang, Yuanyuan Li, Fengmin Huo, Xi Chen, Hui Zhu, Shaochen Guo, Lei Fu, Bin Wang, and Yu Lu

Corresponding Author(s): Yu Lu, Beijing Chest Hospital affiliated to Capital Medical University, Beijing Tuberculosis and Thoracic Tumor Research Institute

Review Timeline:

Submission Date:	January 2, 2023
Editorial Decision:	February 13, 2023
Revision Received:	April 13, 2023
Editorial Decision:	April 24, 2023
Revision Received:	June 13, 2023
Accepted:	July 3, 2023

Editor: Alexandra Aubry

Reviewer(s): The reviewers have opted to remain anonymous.

Transaction Report:

DOI: <https://doi.org/10.1128/spectrum.00002-23>

February 13, 2023

Dr. Yu Lu

Beijing Chest Hospital affiliated to Capital Medical University, Beijing Tuberculosis and Thoracic Tumor Research Institute
Pharmacology
Beijing Tuberculosis and Thoracic Tumor Research Institute
Beijing
China

Re: Spectrum00002-23 (Rv1453, a novel determinant of resistance to clofazimine in *Mycobacterium Tuberculosis*)

Dear Dr. Yu Lu:

Link Not Available

Sincerely,

Alexandra Aubry

Journals Department
Reviewer comments:

Reviewer #1 (Comments for the Author):

Zhang et al. conclude that "a mutation in the intergenic region of the Rv1453 gene confers resistance to clofazimine" and "were crucial in CFZ resistance". However, the data, at best, show that the mutation in question correlates with CFZ resistance. The knockout data are the potentially strongest evidence for involvement of Rv1453 in CFZ resistance. However, I have several major concerns regarding the underlying evidence.

Major comments:

1. Lines 103-104: Presumably you tested a two-fold dilution series from 10 to 0.005 mg/L rather than ug/L? If so, I do not

understand how you get MICs that do not conform to this dilution series (e.g. 3.618 mg/L for 11492) and how you used 1.2 mg/L as the critical concentration, even though this concentration was not tested. In this context, I also noticed that TBI-166, a CFZ analogue, was tested for at least some strains (Table 2). Explain how this was done and comment on the fact that cross-resistance appears not to have been complete.

2. Line 158: "Confirmed" implies that some strains were tested more than once. Please clarify how often the MIC of the 11 clinical resistant strains was measured and what measures for quality control (QC) were taken (e.g. whether H37Rv was included whenever these were tested and, if so, include the H37Rv MICs). The same issues apply to the MICs in Table 4, which, in fact, do not match what you wrote in lines 222-223. It is also not clear whether you constructed a single or four knockouts as the title of Table 4 mentions "knockouts" but "strain" is used in the table itself. Are the results for H37Rv and a single knockout tested four times?

3. Line 169: What mutation frequency did you use as a cut-off to call mutations? It is known that low-frequency heteroresistance plays a major role for Rv0678, which means that it is important for you to check this. Also, please note that the region upstream of Rv1453 is annotated as a repeat region in H37Rv ("1260 bp imperfect direct repeat 1, second copy at 1633531..1634790", which is upstream of *ctaB*). Therefore, make sure that the inter-genic mutation that you called is not an artefact and that this did not affect your knockout experiment either.

4. I am concerned that the promoter mutation may just be associated with CFZ resistance rather than causative. Please construct a phylogenetic tree of all 11 resistant and 7 susceptible strains. Is the promoter mutation mono-phyletic (if so, how closely related are they?) or has it evolved repeatedly (i.e. is homoplastic), which would be a signal for positive selection and potentially a signal for drug-resistance? In fact, you imply independent acquisition of resistance in line 183 ("each of the genes or Rv1453 intergenomic region could be mutated individually") without providing any evidence, unless I misunderstood what you were trying to say. In this context, please check whether phenotypic DST results are in line with the genotypic predictions based on the WGS data. Similarly, do you have any typing information for the additional strains from Table 3 (e.g. spoligotyping or MIRU-VNTR)? Why did you not sequence the known resistance genes for these strains? Sequencing their genomes would provide stronger evidence and allow for them to be included in the above phylogenetic analysis.

5. Why did you not sequence the genome of your knockout(s)? Unexpected artefacts have confounded similar experiments in the past (<https://pubmed.ncbi.nlm.nih.gov/33367727/>)? In my view, simple PCR confirmation is not conclusive and WGS should be carried out.

6. Given the partial cross-resistance with BDQ, I do not understand why you did not measure any BDQ MICs, particularly given that you even checked the patient records for BDQ use.

7. Line 305: "We speculate that the C-17T mutation in the Rv1453 intergenic region alters the Rv1453 protein structure, resulting in the inability of the Rv1453 protein to bind to the target gene." I do not understand how an inter-genic mutation can change the protein structure. Assuming this mutation indeed confers CFZ resistance and that your knockout experiments are sound, would it not make more sense for the mutation to result in resistance by down-regulating Rv1453? Please analyse the inter-genic region for a promoter region and check whether the mutation is likely to make the promoter stronger or weaker.

Minor comments:

1. Throughout the manuscript, including the reference, please correct the capitalizations and italicizations where relevant. Moreover, spaces are either often missing or unnecessary.
2. Line 72: Include *pepQ* as the alternative gene name for Rv2535c and use it throughout the manuscript instead of Rv2535c.
3. The discussion would benefit from being shortened as it contains unnecessary information.
4. Lines 236-237: The 95% binomial confidence interval for your 7% resistance frequency is 3-14%. So, you cannot conclude that your frequency is higher than in the other studies.
5. Line 285: Ismail et al. used MGIT for CFZ MIC testing, which cannot be compared to MABA. Moreover, WHO has now endorsed 1 mg/L as the CC for MGIT. So, this is an inappropriate comparison.
6. I could not find the genome sequences at the SRA using PRJNA914083.

Reviewer #2 (Comments for the Author):

This manuscript describes microbiological and genomics experiments to establish an association between a SNP in the intergenic region upstream of Rv1453 and resistance to clofazimine (CFZ). In a prior publication (ref 32), the authors reported a mutation frequency of 54.5% (6/11) in the Rv1453 intergenic region of 11 CFZ-resistant clinical isolates and went on to show that an Rv1453 knockout strain had a 4x increase in CFZ and that Rv1453 is a transcriptional repressor of *qor* and Rv1455. In the current study, the authors appear to be describing the same 11 CFZ-resistant isolates mentioned in the prior study and providing the description of the actual mutation (C-17T) and the CFZ MICs against the isolates. They also expand the number to CFZ-resistant isolates evaluated by adding 30 more isolates, 27 (90%) of which have the same mutation. They describe creating an Rv1453 knockout and testing the effect on CFZ MIC, but that appears to have already been reported in their prior study (ref 32). So, the only real novelty to the current manuscript is that the previously mentioned intergenic mutation is now actually described and was demonstrated in 27 additional CFZ-resistant clinical isolates. Although the work appears technically sound, these are very incremental advances. Additional major and minor comments are provided below.

Major comments

The title is too general and implies the findings reported here are more novel than they are since the authors already reported Rv1453 as a novel determinant of CFZ resistance. A title like "Single nucleotide polymorphism upstream of Rv1453 is

associated with clofazimine resistance in *Mycobacterium tuberculosis*" would be more specific and appropriate.

Lines 39 and 86: "confers" should be changed to "is associated with" since the authors did not prove the mutation confers resistance.

The authors need to cite their prior study (ref 32) in the Introduction and clarify whether the 11 isolates described here are the same 11 isolates described in the Introduction of the prior paper. If they are, then the authors need to make it clear that the current manuscript is providing more detail on those 11 isolates that were previously described and that prompted their previous study on Rv1453. They may also add that they studied 30 additional CFZ-resistant isolates for the presence of the mutation.

A major weakness of the study is that the presence or absence of the C-17T mutation is only described in CFZ-resistant isolates and not CFZ-susceptible isolates. There is no explicit statement confirming the absence of the mutation in the 7 CFZ-susceptible strains that underwent WGS. It would also be helpful to look for the mutation in additional genomes from CFZ-susceptible strains that they or others have sequenced (for example, from ref 11). Finally, a phylogenomic analysis like that performed in reference 11 would be very helpful to help address the question of whether the C-17T mutation is a phylogenetic marker.

Assuming the construction, verification and testing of the Rv1453 knockout strain is the same as what was done in the prior study (ref 32), these parts of the methods and results (sections 2.7, 3.3-3.5) should be deleted in the current manuscript and the authors should simply cite the prior study showing that knockout was associated with higher MIC.

A figure on the MIC distribution of all 100 isolates mentioned on line 156 should be provided.

Future studies proposed in the Discussion could include recreating the C-17T mutation in a wild-type background to determine if it truly causes resistance and to evaluate its effect on Rv1453 expression.

Minor comments

Line 35: *Mycobacterium*

Lines 71 and 273: suggest changing "major" to "previously described". The authors might also note that the significance of Rv1979c mutations remains unclear (the review by Kadura et al *J Antimicrob Chemother* 2020; 75: 2031-2043 could be cited here).

Line 128: presumably the PCR and sequencing also captured the intergenic region. The region amplified by these primers should be described in more detail.

Line 157: please distinguish here or in Table 1 which strains are the 4 preserved isolates, as it appears that at least 2 of these were previously reported (ref 11) and that reference should be cited here.

Line 163: the CFZ MICs against these 7 CFZ-sensitive strains should be provided.

Line 169: the authors should please clarify if they are describing only the frequencies of previously described mutations in these genes or any mutation identified (whether previously reported or not)

Lines 180-1: it should be stated here whether any of the 7 CFZ-susceptible strains had the same mutation in the Rv1453 intergenic region

Lines 197-8: "crucial in" should be replaced with "associated with" or something similar

Line 201: please confirm the source of these 30 additional CFZ-resistant strains. Were they also isolated between 2014-16? Were mutations in other associated genes (Rv0678, etc) excluded or not? Please provide the MIC distribution, if possible.

Line 223: please explain what is meant by the "multienzyme marker"

Line 275: suggest to change "unclarified" to "unidentified"

Line 276: here the authors should revise the wording so as not to imply that Rv1453 is "new" and that they recently reported it.

Line 277: change "leading to" to "associated with"

Lines 305-6: why not also consider that the mutation results in decreased expression of Rv1453, which seems much more likely than an intergenic SNP causing an alteration in Rv1453 protein structure?

Table 2: TBI-166 is included here. Since it is a CFZ analog, it is important to consider whether the C-17T mutation is associated with resistance to it or not. If it is included in Table 2, the authors should describe the breakpoint applied for TBI-166 and confirm whether it was tested against each isolate in the table. It would be helpful to see the distribution of TBI-166 MICs if available.

Alternatively, the authors could omit TBI-166 from the table

Table 2 footnote: clarithromycin

Table 4: please clarify if this is 4 replicates of MIC testing against the same 2 strains

Staff Comments:

Preparing Revision Guidelines

To submit your modified manuscript, log onto the eJP submission site at <https://spectrum.msubmit.net/cgi-bin/main.plex>. Go to

Author Tasks and click the appropriate manuscript title to begin the revision process. The information that you entered when you first submitted the paper will be displayed. Please update the information as necessary. Here are a few examples of required updates that authors must address:

Please return the manuscript within 60 days; if you cannot complete the modification within this time period, please contact me. If you do not wish to modify the manuscript and prefer to submit it to another journal, please notify me of your decision immediately so that the manuscript may be formally withdrawn from consideration by Microbiology Spectrum.

Answer to the Editorial and Reviewers' Comments

Dear Alexandra Aubry.

The authors appreciate the comments and changes suggested by the editors and reviewers. We enclose a new version of the manuscript that has been revised accordingly. Please find below our point-by-point response to these comments and questions.

Yours sincerely,

Yu Lu

Beijing Chest Hospital, Capital Medical University and Beijing Key Laboratory of Drug Resistance Tuberculosis Research, Beijing Tuberculosis and Thoracic Tumor Research Institute, Beijing, China

EDITOR'S COMMENTS TO AUTHORS:

Reviewer comments:

Reviewer #1 (Comments for the Author):

Zhang et al. conclude that "a mutation in the intergenic region of the Rv1453 gene confers resistance to clofazimine" and "were crucial in CFZ resistance". However, the data, at best, show that the mutation in question correlates with CFZ resistance. The knockout data are the potentially strongest evidence for involvement of Rv1453 in CFZ resistance. However, I have several major concerns regarding the underlying evidence.

Major comments:

1. Lines 103-104: Presumably you tested a two-fold dilution series from 10 to 0.005 mg/L rather than ug/L? If so, I do not understand how you get MICs that do not conform to this dilution series (e.g. 3.618 mg/L for 11492) and how you used 1.2 mg/L as the critical concentration, even though this concentration was not tested. In this context, I also noticed that TBI-166, a CFZ analogue, was tested for at least some strains (Table 2). Explain how this was done and comment on the fact that cross-resistance appears not to have been complete.

Respond:

We feel sorry for our carelessness and changed "ug/liter" to "mg/L" (line 109). Thanks for your correction.

We obtain MIC values by detecting fluorescence and performing calculations. The fluorescence was measured at an excitation wavelength of 530 nm and an emission wavelength of 590 nm. The MIC was defined as the lowest concentration eliciting a reduction in fluorescence of $\geq 90\%$ relative to the mean fluorescence of replicate drug-free controls (line 111 to 114). Specific experimental procedures can be referred to reference 14 as well as Collins and Franzblau, 1997 (DOI:10.1128/AAC.41.5.1004). We used 1.2 mg/L as the critical concentration based on the previous study which showed that the tentative MABA breakpoint of 1.2 mg/L is consistent with the breakpoint suggested by the MGIT 960 method (line 121,

ref 11).

The MICs of TBI-166 were tested by the microplate AlamarBlue assay (MABA). Briefly, the final concentrations of TBI-166 ranged from 10 to 0.005 mg/L. Then added 100 µl bacteria (2×10^5 CFU) to wells. The plates were incubated at 37°C. After day 7 of incubation, added 12.5 µl of 20% Tween 80 and 20 µl of alamarBlue to plates. After incubation at 37°C for another 24 h, the fluorescence was measured at an excitation wavelength of 530 nm and an emission wavelength of 590 nm. The MIC was defined as the lowest concentration eliciting a reduction in fluorescence of $\geq 90\%$ relative to the mean fluorescence of replicate drug-free controls (line 107 to line 114). As shown in Table 2, CFZ-resistant mutants harboring mutation in Rv1453 intergenic were partly resistant to TBI-166, which needs further investigation.

2. Line 158: "Confirmed" implies that some strains were tested more than once. Please clarify how often the MIC of the 11 clinical resistant strains was measured and what measures for quality control (QC) were taken (e.g. whether H37Rv was included whenever these were tested and, if so, include the H37Rv MICs). The same issues apply to the MICs in Table 4, which, in fact, do not match what you wrote in lines 222-223. It is also not clear whether you constructed a single or four knockouts as the title of Table 4 mentions "knockouts" but "strain" is used in the table itself. Are the results for H37Rv and a single knockout tested four times?

Respond:

Thank you for your reminder. We have clarified that the 11 clinical resistant strains were tested three times (line 123). The susceptible reference strain H37Rv was included in each batch as a control and the MIC value of CFZ to H37Rv strain was 0.11 mg/L (line 114 to 116).

Thank you for your reminder. Table 4 shows the results for H37Rv and the single knockout strain tested four times. We have revised the manuscript where the presentation is unclear (line 232, line 235 to 236).

3. Line 169: What mutation frequency did you use as a cut-off to call mutations? It is known that low-frequency heteroresistance plays a major role for Rv0678, which means that it is important for you to check this. Also, please note that the region upstream of Rv1453 is annotated as a repeat region in H37Rv ("1260 bp imperfect direct repeat 1, second copy at 1633531..1634790", which is upstream of *ctaB*). Therefore, make sure that the inter-genic mutation that you called is not an artefact and that this did not affect your knockout experiment either.

Respond:

Thank you for your reminder. The low-frequency heteroresistance plays a major role for Rv0678. Currently, researchers have investigated the detection of minor variants in whole genome sequencing data of *Mycobacterium tuberculosis*. Although for the whole genome sequencing in this study, the mutation results were simultaneously processed by HaplotypeCaller, a real-time de novo algorithm-based tool added by GATK, on the bam files of multiple samples (retaining mutation sites with Qual greater than 30), and the resulting vcf files were annotated using snpEFF software. Therefore we did not define a cut-off value for

the mutation frequency of the mutations. But we think this is a very good suggestion.

Thank you for your reminder. The region upstream of Rv1453 is annotated as a repeat region in H37Rv ("1260 bp imperfect direct repeat 1, second copy at 1633531..1634790", which is upstream of *ctaB*). However, based on the following two points we believe that the intergenic mutation of the Rv1453 gene is not an artefact. First, we performed WGS using 150 bp paired-end reads on the 18 *M. tuberculosis* isolates. Second, we performed PCR and sequencing validation of strains such as 12657 (containing the Rv1453 C-17T mutation), and the results showed that the PCR sequence (about 350 bp) of Rv1453 C-17T could completely match the upstream sequence except for the C-17T site of the Rv1453 gene, but not the "1633531..1634790" sequence. So, the mutation in the intergenic region of the Rv1453 gene does exist.

4. I am concerned that the promoter mutation may just be associated with CFZ resistance rather than causative. Please construct a phylogenetic tree of all 11 resistant and 7 susceptible strains. Is the promoter mutation mono-phyletic (if so, how closely related are they?) or has it evolved repeatedly (i.e. is homoplasic), which would be a signal for positive selection and potentially a signal for drug-resistance? In fact, you imply independent acquisition of resistance in line 183 ("each of the genes or Rv1453 intergenomic region could be mutated individually") without providing any evidence, unless I misunderstood what you were trying to say. In this context, please check whether phenotypic DST results are in line with the genotypic predictions based on the WGS data. Similarly, do you have any typing information for the additional strains from Table 3 (e.g. spoligotyping or MIRU-VNTR)? Why did you not sequence the known resistance genes for these strains? Sequencing their genomes would provide stronger evidence and allow for them to be included in the above phylogenetic analysis.

Respond:

Thanks a lot for your valuable comments. In this paper, we suggest that promoter mutation is associated with CFZ resistance in *M. tuberculosis*. And whether the promoter mutation are associated with CFZ pathogenesis warrants further investigation.

We have constructed a phylogenetic tree of all 11 resistant and 7 susceptible strains (line 218 to line 224, Figure 1). The phylogenetic tree shows that the promoter mutation is not mono-phyletic. However, the restricted number of strains may limit this result.

Based on the results of table 1, MTB clinical isolates 12657, 13480, or 14470 have only one of the mutations in the Rv0678, Rv1979c, Rv2535c, and in the promoter region of Rv1453 gene, respectively. For these four genes, each gene can be mutated independently of the other three genes alone. But by reviewer's reminder, we realized that this conclusion could not be obtained simply, so we deleted this ("each of the genes or Rv1453 intergenomic region could be mutated individually") (line 193).

Thank you for your reminder. We have checked the phenotypic DST results are in line with the genotypic predictions based on the WGS data. Based on the previous study, 0.5 mg/L as

the critical concentration of BDQ (DOI:10.1016/j.ebiom.2018.01.005). And it is worth noting that we did not define the CFZ-resistant strains 12657 and 13908 (containing the Rv0678 mutation) as having a BDQ MIC of less than 0.5 mg/L for BDQ resistance.

Thank you for your suggestion. We don't have typing information for the additional strains from Table 3. The strains in Table 3 were selected to verify whether the Rv1453 promoter mutation was associated with CFZ resistance, and therefore, in this part of the experiment, we performed PCR and sequenced the molecules only for the Rv1453 gene promoter region. Nevertheless, we believe that sequencing these strains is a very good recommendation.

5. Why did you not sequence the genome of your knockout(s)? Unexpected artefacts have confounded similar experiments in the past (<https://pubmed.ncbi.nlm.nih.gov/33367727/>)? In my view, simple PCR confirmation is not conclusive and WGS should be carried out.

Respond:

Thank you for your reminder. The author said that they did not successfully construct Rv1783 (V1221G) mutant strain, but artificially introduced a 900 bp fragment in the strain, which resulted in resistance of this strain to ofloxacin. Although we did not perform whole genome sequencing of the Rv1453 gene knockout strain, the following experiments could demonstrate that the Rv1453 gene knockout strain was successfully constructed. First, unlike the point mutant strain, we replaced the Rv1453 gene of *M. tuberculosis* H37Rv with res-hyg-res gene cassette to construct the knockout strain. That means, if we did not successfully construct Rv1453 gene knockout strain, but artificially introduced a fragment in the strain, then the PCR validation result in line 1 should not appear in only one band (Figure S1), but two bands, one at the position shown in line 2 and one at the position shown in line 1. Second, reference 13 used qRT-PCR to analyze Rv1453 gene expression in the Rv1453 knockout strain and found that Rv1453 gene expression decreased, indicating that the Rv1453 knockout strain was successfully constructed.

6. Given the partial cross-resistance with BDQ, I do not understand why you did not measure any BDQ MICs, particularly given that you even checked the patient records for BDQ use.

Respond:

Thank you for your reminder, we have supplemented the drug resistance profiles of the strains to BDQ (Table 2).

7. Line 305: "We speculate that the C-17T mutation in the Rv1453 intergenic region alters the Rv1453 protein structure, resulting in the inability of the Rv1453 protein to bind to the target gene." I do not understand how an inter-genic mutation can change the protein structure. Assuming this mutation indeed confers CFZ resistance and that your knockout experiments are sound, would it not make more sense for the mutation to result in resistance by down-regulating Rv1453? Please analyse the inter-genetic region for a promoter region and check whether the mutation is likely to make the promoter stronger or weaker.

Respond:

We think this is an excellent suggestion. We have rewritten this part according to the

reviewer's suggestion (line 312 to line 313).

Minor comments:

1. Throughout the manuscript, including the reference, please correct the capitalizations and italicizations where relevant. Moreover, spaces are either often missing or unnecessary.

Respond:

As suggested by the reviewer, we have corrected the capitalizations, italicizations, and spaces where relevant.

2. Line 72: Include pepQ as the alternative gene name for Rv2535c and use it throughout the manuscript instead of Rv2535c.

Respond:

As suggested by the reviewer, we used pepQ throughout the manuscript instead of Rv2535c (line 74 etc).

3. The discussion would benefit from being shortened as it contains unnecessary information.

Respond:

We have deleted unnecessary information of the discussion part according to the reviewer's suggestion (line 250 and line 294).

4. Lines 236-237: The 95% binomial confidence interval for your 7% resistance frequency is 3-14%. So, you cannot conclude that your frequency is higher than in the other studies.

Respond:

We think this is an excellent suggestion. We have removed this unreasonable conclusion which can be found in the revised manuscript (line 250).

5. Line 285: Ismail et al. used MGIT for CFZ MIC testing, which cannot be compared to MABA. Moreover, WHO has now endorsed 1 mg/L as the CC for MGIT. So, this is an inappropriate comparison.

Respond:

We agree with the reviewer's assessment and removed the inappropriate comparison (line 294).

6. I could not find the genome sequences at the SRA using PRJNA914083.

Respond:

Thank you for your reminder. We have created a "reviewer" link to share metadata (<https://dataview.ncbi.nlm.nih.gov/object/PRJNA914083?reviewer=18sngrdqk7qheinr6o14n4udjk>).

Reviewer #2 (Comments for the Author):

This manuscript describes microbiological and genomics experiments to establish an

association between an SNP in the intergenic region upstream of Rv1453 and resistance to clofazimine (CFZ). In a prior publication (ref 32), the authors reported a mutation frequency of 54.5% (6/11) in the Rv1453 intergenic region of 11 CFZ-resistant clinical isolates and went on to show that an Rv1453 knockout strain had a 4x increase in CFZ and that Rv1453 is a transcriptional repressor of qor and Rv1455. In the current study, the authors appear to be describing the same 11 CFZ-resistant isolates mentioned in the prior study and providing the description of the actual mutation (C-17T) and the CFZ MICs against the isolates. They also expand the number of CFZ-resistant isolates evaluated by adding 30 more isolates, 27 (90%) of which have the same mutation. They describe creating an Rv1453 knockout and testing the effect on CFZ MIC, but that appears to have already been reported in their prior study (ref 32). So, the only real novelty to the current manuscript is that the previously mentioned intergenic mutation is now actually described and was demonstrated in 27 additional CFZ-resistant clinical isolates. Although the work appears technically sound, these are very incremental advances. Additional major and minor comments are provided below.

Major comments

The title is too general and implies the findings reported here are more novel than they are since the authors already reported Rv1453 as a novel determinant of CFZ resistance. A title like "Single nucleotide polymorphism upstream of Rv1453 is associated with clofazimine resistance in Mycobacterium tuberculosis" would be more specific and appropriate.

Respond:

As suggested by the reviewer, we modified the title to "Rv1453 gene is associated with clofazimine resistance in Mycobacterium tuberculosis" (line 1)

Lines 39 and 86: "confers" should be changed to "is associated with" since the authors did not prove the mutation confers resistance.

Respond:

We think this is an excellent suggestion. We have changed "confers" to "is associated with" (line 38 and line 89).

The authors need to cite their prior study (ref 32) in the Introduction and clarify whether the 11 isolates described here are the same 11 isolates described in the Introduction of the prior paper. If they are, then the authors need to make it clear that the current manuscript is providing more detail on those 11 isolates that were previously described and that prompted their previous study on Rv1453. They may also add that they studied 30 additional CFZ-resistant isolates for the presence of the mutation.

Respond:

We have added the suggested content to the manuscript in the Introduction (line 85 to line 95).

A major weakness of the study is that the presence or absence of the C-17T mutation is only described in CFZ-resistant isolates and not CFZ-susceptible isolates. There is no explicit statement confirming the absence of the mutation in the 7 CFZ-susceptible strains that underwent WGS. It would also be helpful to look for the mutation in additional genomes from

CFZ-susceptible strains that they or others have sequenced (for example, from ref 11). Finally, a phylogenomic analysis like that performed in reference 11 would be very helpful to help address the question of whether the C-17T mutation is a phylogenetic marker.

Respond:

Yes, the C-17T mutation is only present in CFZ-resistant isolates and not in CFZ-susceptible isolates. Line 176-177: "A total of 532 SNV mutant genes and 121 mutation sites in the intergenic region were obtained by removing the detected mutation sites from seven sensitive CFZ strains" has shown that the presence of the C-17T mutation is only in CFZ-resistant isolates. However, our wording may not have been clear enough, so we have amended the manuscript to explicitly confirm the absence of this mutation in the 7 CFZ-susceptible strains that underwent WGS (line 191 to line 192).

We constructed a phylogenomic analysis and supplemented them in the manuscript (line 218 to line 224, Figure 1).

Assuming the construction, verification and testing of the Rv1453 knockout strain is the same as what was done in the prior study (ref 32), these parts of the methods and results (sections 2.7, 3.3-3.5) should be deleted in the current manuscript and the authors should simply cite the prior study showing that knockout was associated with higher MIC.

Respond:

Thank you for your reminder. In fact, the study presented here is prior to ref 13, where we did not detail the construction of the Rv1453 knockout strain, so we believe that this section can be appropriately retained. As well, the knockout strain MIC values were determined making this study more complete and also allowing comparison with previous data.

A figure on the MIC distribution of all 100 isolates mentioned on line 156 should be provided. Future studies proposed in the Discussion could include recreating the C-17T mutation in a wild-type background to determine if it truly causes resistance and to evaluate its effect on Rv1453 expression.

Respond:

The figure shows MIC distribution of CFZ for all 100 isolates. We detected the MIC values of all 100 isolates in order to obtain CFZ-resistant clinical isolates, and our group has previously reported MIC distribution of clinical isolates (ref 11).

We have added the suggested content to the manuscript in the discussion (line 315 to line 317).

Minor comments

Line 35: Mycobacterium

Respond:

We feel sorry for our carelessness. In our resubmitted manuscript, the typo is revised (line 34). Thanks for your correction.

Lines 71 and 273: suggest changing "major" to "previously described". The authors might also note that the significance of Rv1979c mutations remains unclear (the review by Kadura et al J Antimicrob Chemother 2020; 75: 2031-2043 could be cited here).

Respond:

As suggested by the reviewer, we have changed "major" to "previously described" (line 74 and line 286) and added this references to support the idea that the significance of Rv1979c mutations remains unclear (line 275 to line 276, ref 25).

Line 128: presumably the PCR and sequencing also captured the intergenic region. The region amplified by these primers should be described in more detail.

Respond:

We have added the suggested content to the manuscript (line 211 to line 213).

Line 157: please distinguish here or in Table 1 which strains are the 4 preserved isolates, as it appears that at least 2 of these were previously reported (ref 11) and that reference should be cited here.

Respond:

We have added the suggested content to the manuscript in Table 1.

Line 163: the CFZ MICs against these 7 CFZ-sensitive strains should be provided.

Respond:

We think this is an excellent suggestion. Line 169 to line 170, the CFZ MICs against these 7 CFZ-sensitive strains were added.

Line 169: the authors should please clarify if they are describing only the frequencies of previously described mutations in these genes or any mutation identified (whether previously reported or not)

Respond:

We have added the suggested content to the manuscript (line 179).

Lines 180-1: it should be stated here whether any of the 7 CFZ-susceptible strains had the same mutation in the Rv1453 intergenic region

Respond:

We have stated in the manuscript that this mutation is not present in the 7 CFZ-susceptible

strains (line 191 to 192)

Lines 197-8: "crucial in" should be replaced with "associated with" or something similar

Respond:

We think this is an excellent suggestion. We have changed "crucial in" to "related to" (line 206).

Line 201: please confirm the source of these 30 additional CFZ-resistant strains. Were they also isolated between 2014-16? Were mutations in other associated genes (Rv0678, etc) excluded or not? Please provide the MIC distribution, if possible.

Respond:

The 30 additional strains were isolated between 2013-2018. Because we wanted to verify the Rv1453 intergenic region mutation in CFZ-resistant clinical strains, we did not test other genes. These 30 CFZ-resistant clinical isolates all had MIC value greater than 1.2 mg/L, so we did not plot their MIC distributions.

Line 223: please explain what is meant by the "multienzyme marker"

Respond:

We sincerely thank the reviewer for careful reading. We have changed the "multienzyme marker" to "multifunctional microplate reader" (line 238).

Line 275: suggest to change "unclarified" to "unidentified"

Respond:

We think this is an excellent suggestion. We have changed "unclarified" to "unidentified" (line 288).

Line 276: here the authors should revise the wording so as not to imply that Rv1453 is "new" and that they recently reported it.

Respond:

As suggested by the reviewer, we have changed "a new" to "the" (line 289).

Line 277: change "leading to" to "associated with"

Respond:

We think this is an excellent suggestion. We have changed "leading to" to "associated with" (line 290).

Lines 305-6: why not also consider that the mutation results in decreased expression of Rv1453, which seems much more likely than an intergenic SNP causing an alteration in Rv1453 protein structure?

Respond:

We think this is an excellent suggestion. We have rewritten this part according to the Reviewer's suggestion (line 312 to line 313).

Table 2: TBI-166 is included here. Since it is a CFZ analog, it is important to consider

whether the C-17T mutation is associated with resistance to it or not. If it is included in Table 2, the authors should describe the breakpoint applied for TBI-166 and confirm whether it was tested against each isolate in the table. It would be helpful to see the distribution of TBI-166 MICs if available. Alternatively, the authors could omit TBI-166 from the table

Respond:

At present, there is no recognized cut-off value for TBI-166. Therefore, the 10-fold MIC value of TBI-166 against H37Rv, that is to say, 0.6 mg/L, is tentatively set as the cut-off value of TBI-166. And we have described the breakpoint applied for TBI-166 (line 121 to 123). We determined the MIC value of TBI-166 for each strain in table 2. However, there are only 11 strains and TBI-166 was not the focus of this study, we did not map the distribution of TBI-166 MICs.

Table 2 footnote: clarithromycin

Respond:

We feel sorry for our carelessness. In our resubmitted manuscript, the typo is revised (Table 2). Thanks for your correction.

Table 4: please clarify if this is 4 replicates of MIC testing against the same 2 strains

Respond:

We think this is an excellent suggestion. We have clarified this is 4 replicates of MIC testing against the same 2 strains (line 235 to line 236).

April 24, 2023

Dr. Yu Lu

Beijing Chest Hospital affiliated to Capital Medical University, Beijing Tuberculosis and Thoracic Tumor Research Institute
Pharmacology
Beijing Tuberculosis and Thoracic Tumor Research Institute
Beijing
China

Re: Spectrum00002-23R1 (*Rv1453 gene is associated with clofazimine resistance in Mycobacterium tuberculosis*)

Dear Dr. Yu Lu:

Link Not Available

Sincerely,

Alexandra Aubry

Journals Department
Reviewer comments:

Reviewer #1 (Comments for the Author):

The changes by Zhang et al. have improved the manuscript, but further changes are needed.

Major comments:

1. Please combine Tables 1 and 2 as a single Supplementary Excel table and add the following information for the 104 MDR-TB strains initially analyzed (for some strains you will obviously not have some of the information, such as the genome or the medication):

a. ID

- b. Date of isolation
 - c. Do not show the DST profile as a list of resistances as one cannot tell whether, for example, 12657 was susceptible to BDQ or simply not tested. Instead, add a column for each drug with the method and CC tested and indicate whether the strain was susceptible, resistant or not tested. If the MIC for any other drug was tested (e.g. BDQ) include that as well.
 - d. All CFZ MICs.
 - e. The genome accession for each strain.
 - f. The Coll typing number based on <https://pubmed.ncbi.nlm.nih.gov/25176035/>
 - g. Rv0678, Rv1979c, pepQ, Rv1453 AND atpE mutations based on WGS.
 - h. Medication record
 - i. CFZ treatment
 - j. A column to show whether the strain was part of your earlier study (<https://pubmed.ncbi.nlm.nih.gov/28320727/>).
2. In the response to review 2, you suggest that some of the CFZ MICs were previously published (<https://pubmed.ncbi.nlm.nih.gov/28320727/>). So, why does your current study not feature the CFZ-R strain 10601?
 3. You provide no information about the additional 30 CFZ-R strains analysed in section 3.3. Convert Table 3 to a new supplementary table and provide relevant information for them (see point 1 regarding the set of 104 MDR-TB strains).
 4. I would suggest that you get input from someone who has more experience in phylogenetic analyses than me to improve the tree. In my view, at least the following analysis are needed:
 - a. Highlight which strains are susceptible/resistant to CFZ/BDQ/TBI-166 and which ones have the Rv1453 mutation.
 - b. Distinguish the different lineages and root the tree appropriately and clearly state whether there is evidence of homoplasy (i.e. the independent evolution of the Rv1453 mutation).
 - c. Explain what the scale corresponds to.
 5. In the discussion, please comment explicitly on whether the clinical DST results from your study suggest that Rv1453 may confer cross-resistance to BDQ and/or TBI-166.

Minor comments:

1. There is no need to include "gene" in the title.
2. Round percentages to the nearest integer (e.g. 55% in line 28) given the small sample size.
3. Line 28: Clarify that the 30 CFZ-R strains represent an independent set of strains to the 11 discussed earlier.
4. In the introduction, mention that atpE is the target of BDQ and, thus, is not involved in CFZ resistance. Also, mention that the role of Rv1979c in CFZ resistance has been questioned (<https://pubmed.ncbi.nlm.nih.gov/32143680/>).
5. Line 146: It is not clear whether the knockout in this study is the same as in your previous study (<https://pubmed.ncbi.nlm.nih.gov/34594117/>).
6. Correct line 188 given that one of the CFZ-S strains is H37Rv and, therefore, not a clinical strain.
7. Line 189: You cannot write "highly associated" without running an appropriate statistical analysis. I would change it to "potentially associated."
8. Lines 205-6 are repetitive and can be deleted.
9. Move section 3.4 before 3.3 as it covers the strains from section 3.2.
10. Move section 3.5 to the supplement.
11. The discussion repeats information from the introduction and results unnecessarily and needs to be shortened.
12. Line 336: The main limitation is the small number of CFZ-S isolates analysed, particularly by WGS.
13. Contrary to the authors' rebuttal, they have not corrected the capitalizations, italicizations etc. In fact, the e-pages of some articles are missing (e.g. <https://pubmed.ncbi.nlm.nih.gov/28739779/>) and I do not understand why reference 32 has a star in its title.

Reviewer #2 (Comments for the Author):

The authors responses to the prior reviews are mostly satisfactory. A few issues persist, as described in the comments below. Abstract: with the exception of this short phrase on lines 28-29: "Among 30 clofazimine-resistant clinical 28 isolates, 27 had mutations in the intergenic region of the Rv1453 gene", the entire abstract describes experiments and results previously reported in the introduction and/or the methods and results of the authors' prior publication (reference # 13). Of the entire abstract, the only new result is the finding of the same mutation in 27 of 30 additional CFZ-resistant strains tested. The authors should rewrite the abstract to make it very clear what was previously reported and what new information is being presented now. For example, they previously reported a mutation or mutations in the intergenic region of Rv1453 associated with CFZ resistance and went on to show that an Rv1453 knockout had an elevated CFZ MIC; and now in the present paper they are going back and describing in detail how the mutation was identified and further exploring its association with CFZ resistance by testing additional isolates.

Line 37: suggest revising to "in this study, we found additional evidence that a mutation in the intergenic region...". Similar to the above comment, the authors should not suggest that the intergenic mutation is being reported for the first time here.

Line 95: "this study provides further evidence..."

Line 281: suggest revising to "pepQ encodes a putative proline-specific..."

Lines 289-90: suggest revising to "Here, we identify a mutation in the intergenic region of..." since the gene's association with resistance was already identified in their previous paper.

Lines 303-307: suggest revising to "We previously determined that the Rv1453 gene is associated with CFZ resistance in M.

tuberculosis by constructing Rv1453 knockout strains and testing the CFZ MIC values. In the present study, we provide further evidence that the C-17T mutation in the intergenic region of the Rv1453 gene is associated with CFZ resistance by expanding the number of clinical isolates screened." Or something similar.

Sections 2.7, 3.5 and 3.6 are reporting methods and results already reported in the same or similar detail in their prior paper and absolutely should not be presented again as if they are new.

Staff Comments:

Preparing Revision Guidelines

Please return the manuscript within 60 days; if you cannot complete the modification within this time period, please contact me. If you do not wish to modify the manuscript and prefer to submit it to another journal, please notify me of your decision immediately so that the manuscript may be formally withdrawn from consideration by Microbiology Spectrum.

Answer to the Editorial and Reviewers' Comments

Dear Alexandra Aubry.

The authors appreciate the comments and changes suggested by the editors and reviewers. We enclose a new version of the manuscript that has been revised accordingly. Please find below our point-by-point response to these comments and questions.

Yours sincerely,

Yu Lu

Beijing Chest Hospital, Capital Medical University and Beijing Key Laboratory of Drug Resistance Tuberculosis Research, Beijing Tuberculosis and Thoracic Tumor Research Institute, Beijing, China

Reviewer #1 (Comments for the Author):

The changes by Zhang et al. have improved the manuscript, but further changes are needed.

Major comments:

1. Please combine Tables 1 and 2 as a single Supplementary Excel table and add the following information for the 104 MDR-TB strains initially analyzed (for some strains you will obviously not have some of the information, such as the genome or the medication):

a. ID

b. Date of isolation

c. Do not show the DST profile as a list of resistances as one cannot tell whether, for example, 12657 was susceptible to BDQ or simply not tested. Instead, add a column for each drug with the method and CC tested and indicate whether the strain was susceptible, resistant or not tested. If the MIC for any other drug was tested (e.g. BDQ) include that as well.

d. All CFZ MICs.

e. The genome accession for each strain.

f. The Coll typing number based on <https://pubmed.ncbi.nlm.nih.gov/25176035/>

g. Rv0678, Rv1979c, pepQ, Rv1453 AND atpE mutations based on WGS.

h. Medication record

i. CFZ treatment

j. A column to show whether the strain was part of your earlier study (<https://pubmed.ncbi.nlm.nih.gov/28320727/>).

Respond:

Thank you for your suggestion. We have added id、 date of isolation、 DST profile of BDQ and TBI-166、 SRA (the genome accession for each strain) 、 lineage、 preserved isolates or not of the 11 CFZ-resistant clinical isolates (Table 1), and id、 date of isolation、 medication administration record、 drug resistance profile, and CFZ treatment or not of CFZ-sensitive MDR-TB strains (Table S1).

2. In the response to review 2, you suggest that some of the CFZ MICs were previously published (<https://pubmed.ncbi.nlm.nih.gov/28320727/>). So, why does your current study not

feature the CFZ-R strain 10601?

Respond:

For whole-genome sequencing, the genomic DNA of strain 10601 needed to be extracted, and the DNA amounts did not meet the standards for whole-genome sequencing because the strain grew poorly, so it was not included.

3. You provide no information about the additional 30 CFZ-R strains analysed in section 3.3. Convert Table 3 to a new supplementary table and provide relevant information for them (see point 1 regarding the set of 104 MDR-TB strains).

Respond:

As shown in Table S2, we have added id, date of isolation, medication administration record and drug resistance profile, CFZ treatment or not of the 30 CFZ-resistant clinical isolates used for validation.

4. I would suggest that you get input from someone who has more experience in phylogenetic analyses than me to improve the tree. In my view, at least the following analysis are needed:

a. Highlight which strains are susceptible/resistant to CFZ/BDQ/TBI-166 and which ones have the Rv1453 mutation.

b. Distinguish the different lineages and root the tree appropriately and clearly state whether there is evidence of homoplasy (i.e. the independent evolution of the Rv1453 mutation).

c. Explain what the scale corresponds to.

Respond:

We annotated phylogenetic tree of which strains are susceptible/resistant to CFZ/BDQ/TBI-166 and which ones have the Rv1453 mutation and explained the scale corresponds to genetic distance (Figure 1, line 199-200). We rooted the tree with H37Rv. Phylogenetic tree shows that there is evidence of homoplasy, and the Rv1453 mutation has independent evolved four times (line 201-202).

5. In the discussion, please comment explicitly on whether the clinical DST results from your study suggest that Rv1453 may confer cross-resistance to BDQ and/or TBI-166.

Respond:

We clearly state that the MIC results from this study indicate that Rv1453 may confer cross-resistance to TBI-166 in the discussion (line 284-285).

Minor comments:

1. There is no need to include "gene" in the title.

Respond:

We have deleted "gene" in the title (line 1).

2. Round percentages to the nearest integer (e.g. 55% in line 28) given the small sample size.

Respond:

We used 55% throughout the manuscript instead of 54.5% (line 29, 307).

3. Line 28: Clarify that the 30 CFZ-R strains represent an independent set of strains to the 11 discussed earlier.

Respond:

We have clarified that the 30 CFZ-R strains did not belong to 100 MDR-TB strains as "and validates mutation in the intergenic region of Rv1453 gene in a further 30 CFZ-resistant clinical isolates which not belongs to 100 MDR-TB isolates" (line 95).

4. In the introduction, mention that *atpE* is the target of BDQ and, thus, is not involved in CFZ resistance. Also, mention that the role of Rv1979c in CFZ resistance has been questioned (<https://pubmed.ncbi.nlm.nih.gov/32143680/>).

Respond:

We have mentioned that "*atpE* is the target of BDQ and, thus, is not involved in CFZ resistance." and "The significance of Rv1979c mutations remains unclear (8)." in the introduction. (line 75-77)

5. Line 146: It is not clear whether the knockout in this study is the same as in your previous study (<https://pubmed.ncbi.nlm.nih.gov/34594117/>).

Respond:

Yes, the knockout strain in this study is the same as previous study, and we have removed this section to avoid duplication with the previously published article (line 148).

6. Correct line 188 given that one of the CFZ-S strains is H37Rv and, therefore, not a clinical strain.

Respond:

We have corrected this sentence to " 11 CFZ-resistant clinical strains and 7 CFZ-sensitive strains " which indicated that H37Rv is not a clinical isolate (line 179-180).

7. Line 189: You cannot write "highly associated" without running an appropriate statistical analysis. I would change it to "potentially associated."

Respond:

We used "potentially associated" instead of "highly associated"(line 180).

8. Lines 205-6 are repetitive and can be deleted. Respond:

We have deleted this sentence "Whole-genome sequencing results showed that mutations in the Rv1453 intergenic region were related to CFZ resistance. " (line 195).

9. Move section 3.4 before 3.3 as it covers the strains from section 3.2.

Respond:

We have moved section 3.4 before 3.3 (line 196-203). Now section 3.3 is "Phylogenetic analysis of CFZ-resistant isolates", section 3.4 is "Validation of the Rv1453 intergenic region mutation in CFZ-resistant clinical strains".

10. Move section 3.5 to the supplement.

Respond:

We have deleted section 3.5 because it was same as previous study. In the discussion section, we cited the MIC results previously reported (line 203).

11. The discussion repeats information from the introduction and results unnecessarily and needs to be shortened.

Respond:

We have deleted repeats information to shortened discussion. Line 234 "Resistance-associated mutations in the Rv0678 gene, which encodes a transcriptional regulator of the MmpS5-MmpL5 efflux pump, increase the MIC of CFZ by 2- to 4-fold (8, 10, 20). Thus, ". Line 246 "Mutations in Rv1979c and pepQ have also been reported to be associated with CFZ resistance in M. tuberculosis (9, 10). The significance of Rv1979c mutations remains unclear (25)". Line 288" Similar to the Rv1453 gene, mutations in the Rv0678, Rv1979c, and pepQ genes were associated with modest 4-fold or less than 4-fold increases in MICs (25) ".

12. Line 336: The main limitation is the small number of CFZ-S isolates analysed, particularly by WGS.

Respond:

We have added this as the main limitation "First, the main limitation of our study is the small number of CFZ-resistant and CFZ-sensitive isolates analyzed, particularly by WGS. " (line 299).

13. Contrary to the authors' rebuttal, they have not corrected the capitalizations, italicizations etc. In fact, the e-pages of some articles are missing (e.g. <https://pubmed.ncbi.nlm.nih.gov/28739779/>) and I do not understand why reference 32 has a star in its title.

Respond:

We have corrected the capitalizations (line 347, line 376-382, line 412, line 414, line 416,), italicizations (line 353, line 366, line 368, line 372, line 374, line 380, line 382, line 402, line 416, line 422, line 426, line 428, line 433, line 435) and e-pages (line 380, line 413) in reference. There was a star in reference 32's title (<https://www.sciencedirect.com/science/article/abs/pii/B978012801238301895X>).

Reviewer #2 (Comments for the Author):

The authors responses to the prior reviews are mostly satisfactory. A few issues persist, as described in the comments below.

Abstract: with the exception of this short phrase on lines 28-29: "Among 30 clofazimine-resistant clinical 28 isolates, 27 had mutations in the intergenic region of the Rv1453 gene", the entire abstract describes experiments and results previously reported in the introduction and/or the methods and results of the authors' prior publication (reference # 13). Of the entire abstract, the only new result is the finding of the same mutation in 27 of 30 additional CFZ-resistant strains tested. The authors should rewrite the abstract to make it very clear what was previously reported and what new information is being presented now. For

example, they previously reported a mutation or mutations in the intergenic region of Rv1453 associated with CFZ resistance and went on to show that an Rv1453 knockout had an elevated CFZ MIC; and now in the present paper they are going back and describing in detail how the mutation was identified and further exploring its association with CFZ resistance by testing additional isolates.

Respond:

We have rewritten the abstract as "...We previously reported a mutation located in the intergenic region of *Rv1453* that was linked to resistance to CFZ, and demonstrated that an *Rv1453* knockout resulted in an increased minimum inhibitory concentration (MIC) of CFZ. The current study aims to go back and describe in detail how the mutation was identified and further explore its association with CFZ resistance by testing additional 30 isolates..." (line 21-25, line 31-32).

Line 37: suggest revising to "in this study, we found additional evidence that a mutation in the intergenic region...". Similar to the above comment, the authors should not suggest that the intergenic mutation is being reported for the first time here.

Respond:

We have revised the sentence to "in this study, we review and detail the findings of the mutation of intergenic region of Rv1453 and found additional evidence that this mutation related to clofazimine resistance in 30 additional isolates." (line 38-40).

Line 95: "this study provides further evidence..."

Respond:

We have revised the sentence to " This study provides further evidence that the Rv1453 gene and CFZ resistance are related." (line 97-98)

Line 281: suggest revising to "pepQ encodes a putative proline-specific..."

Respond:

We have revised to "pepQ encodes a putative proline-specific aminopeptidase, but the mechanism by which pepQ mutations result in reduced susceptibility to CFZ is unclear." (line 250-251)

Lines 289-90: suggest revising to "Here, we identify a mutation in the intergenic region of..." since the gene's association with resistance was already identified in their previous paper.

Respond:

We have revised to "Here, we identified a mutation in the intergenic region of Rv1453 (C-17T) was associated with CFZ resistance" (line 258-259)

Lines 303-307: suggest revising to "We previously determined that the Rv1453 gene is associated with CFZ resistance in *M. tuberculosis* by constructing Rv1453 knockout strains and testing the CFZ MIC values. In the present study, we provide further evidence that the C-17T mutation in the intergenic region of the Rv1453 gene is associated with CFZ resistance by expanding the number of clinical isolates screened." Or something similar.

Respond:

We have revised to "We previously determined that the Rv1453 gene is associated with CFZ resistance in *M. tuberculosis* by constructing Rv1453 knockout strains and testing the CFZ MIC values."(line 272-274)

Sections 2.7, 3.5 and 3.6 are reporting methods and results already reported in the same or similar detail in their prior paper and absolutely should not be presented again as if they are new.

Respond:

Thank you for your suggestion. We have deleted sections 2.7, 3.5 and 3.6 which have been reported in the prior paper (line 148, line 203).

July 3, 2023

Dr. Yu Lu
Beijing Chest Hospital affiliated to Capital Medical University, Beijing Tuberculosis and Thoracic Tumor Research Institute
Pharmacology
Beijing Tuberculosis and Thoracic Tumor Research Institute
Beijing
China

Re: Spectrum00002-23R2 (*Rv1453 is associated with clofazimine resistance in Mycobacterium tuberculosis*)

Dear Dr. Yu Lu:

Your manuscript has been accepted, and I am forwarding it to the ASM Journals Department for publication. You will be notified when your proofs are ready to be viewed.

The following minor comments made by one of the reviewers should be corrected in the published version:

Minor comments:

1. Introduce "BDQ" as an abbreviation in line 75 instead of 77.
2. Your methods sections only mention the initial set of 100 MDR-TB strains, but based on line 158 you identified 4 CFZ-R strains in addition to your validation set of 30 CFZ-R strains. Please rewrite the method to include all strains and cross-reference Table S1 and S2.
3. Use "susceptible" instead of "sensitive" throughout the manuscript, including the figures.
4. Italicize "in vitro" through the manuscript.

Sincerely,

Alexandra Aubry
Editor, Microbiology Spectrum
